# FPGA Implementation for Elliptic Curve Cryptography Algorithm and Circuit with High Efficiency and Low Delay for IoT Applications

**DOI:** 10.3390/mi14051037

**Published:** 2023-05-12

**Authors:** Deming Wang, Yuhang Lin, Jianguo Hu, Chong Zhang, Qinghua Zhong

**Affiliations:** 1School of Electronics and Information Engineering, South China Normal University, Foshan 528225, China; is04wdm@mail3.sysu.edu.cn; 2Development Research Institute of Guangzhou Smart City, Guangzhou 510805, China; hujguo@mail.sysu.edu.cn; 3School of Physics and Telecommunication Engineering, South China Normal University, Guangzhou 510006, China; 2020022244@m.scnu.edu.cn; 4School of Microelectronics Science and Technology, Sun Yat-sen University, Zhuhai 519082, China; zhangch366@mail2.sysu.edu.cn

**Keywords:** elliptic curve cryptography (ECC), field-programmable gate array (FPGA), Internet of Things (IoT), point multiplication, Montgomery reduction

## Abstract

The Internet of Things requires greater attention to the security and privacy of the network. Compared to other public-key cryptosystems, elliptic curve cryptography can provide better security and lower latency with shorter keys, rendering it more suitable for IoT security. This paper presents a high-efficiency and low-delay elliptic curve cryptographic architecture based on the NIST-$p_{256}$ prime field for IoT security applications. A modular square unit utilizes a fast partial Montgomery reduction algorithm, demanding just a mere four clock cycles to complete a modular square operation. The modular square unit can be computed simultaneously with the modular multiplication unit, consequently improving the speed of point multiplication operations. Synthesized on the Xilinx Virtex-7 FPGA platform, the proposed architecture completes one PM operation in 0.08 ms using 23.1 k LUTs at 105.3 MHz. These results show significantly better performance compared to that in previous works.

## 1. Introduction

IoT security faces numerous challenges regarding sensitive issues such as device authentication, software vulnerabilities, and privacy leaks. As IoT technology advances, and the number of smart terminal devices increases, these security concerns become increasingly pressing. Elliptic curve cryptography (ECC) has been well-suited for use in IoT security. Compared to other cryptographic systems, ECC achieves better security levels and lower latency with shorter key lengths. Independently introduced by Koblitz [1] and Miller [2] in 1985, ECC has since been embraced and standardized by international organizations such as ANSI [3], IEEE [4], NIST [5], and SCA [6]. ECC has found extensive use in IoT applications, including NFC [7], RFID [8], and DTLS [9].

Elliptic curve cryptography can be realized through software [10] and hardware [11] approaches. Software-based ECC offers low design costs and high portability, but it is constrained by processor performance and lower efficiency, rendering it unsuitable for high-performance IoT applications. Hardware implementations, including field-programmable gate arrays (FPGAs) and application-specific integrated circuits (ASICs), provide distinct advantages. ASIC-based ECC excels in encryption/decryption speed, security strength, hardware resource utilization, and power consumption. FPGA-based ECC leverages reprogrammability, reconfigurability, compatibility, and parallel computing to achieve reduced latency and enhanced throughput.

Since point multiplication (PM) is the major and critical operation during the whole encryption process, most researchers focus on optimizing the PM hardware architectures to improve the overall performance of ECC. To accelerate the point multiplication process, the modular multiplier has been widely studied for its high logic complexity and long latency. The Radix-4 algorithm proposed in [12] improves the serial interleaved multiplication to achieve a smaller area and higher frequency, which is particularly useful in low-power and resource-constrained situations. In [13], the combination of a single-cycle full-precision multiplier and a fast-reduction algorithm was implemented to achieve extremely high speed. The processor in [14] uses the Karatsuba–Ofman multiplication and fast-reduction algorithm to perform a trade-off between the area and performance of ECC. In [15], the Toom–Cook algorithm was used to improve the multiplier, further reducing multiplication complexity. In [16,17], the fast partial Montgomery reduction algorithm achieved better performance and a more balanced area. In contrast to the majority of works centered on modular multiplication, the authors in [18] presented a low-complexity modular squaring scheme to decrease the cycle time required for point multiplication. Additionally, some studies focused on binary field curves [19,20], but the NIST-*p*_256_ curve is a better choice because it is a widely adopted and standardized elliptic curve, offering strong security and compatibility with various cryptographic protocols. Its performance and smaller keys render it an attractive choice compared to other curves.

Existing works have conducted extensive research on the main frequency, delay, power consumption, and complexity of elliptic curve cryptography circuits. However, the high hardware-resource requirements and long computational delay remain significant challenges. This paper aims to design a high-efficiency and low-delay ECC point multiplication architecture with NIST prime curve $p_{256}=2^{256}-2^{224}+2^{192}+2^{96}-1$. The main contributions of this paper are as follows:A modular square unit using a fast partial Montgomery reduction algorithm is proposed to significantly reduce the area. It takes only four clock cycles to complete a modular square operation. Modular multiplication and modular square operations can be computed in parallel to achieve high speed.A modified point addition (PA) operation and a modified point doubling (PD) operation are proposed. It takes 21 cycles to compute a PD and 32 cycles to compute a PA. The number of operation clock cycles is reduced to the minimum.This paper proposes a high-speed modular inversion algorithm based on the extended Euclidean algorithm, but some modifications were performed using the two modular adders. It takes about 300 clock cycles to perform a modular inverse operation, which is only 83% of the Radix-2 algorithm.

The rest of this paper is organized as follows. Section 2 introduces the background knowledge about ECC and the point multiplication algorithm. Section 3 shows the hardware implementation of the ECC architecture. Section 4 gives a comparison of the performance of the FPGA structure with other works. Section 5 summarizes this work. For a detailed explanation of the abbreviations, please refer to Appendix A.

## 2. Preliminaries

### 2.1. Elliptic Curve Cryptograph

An elliptic curve *E* over a prime field is usually defined with the Weierstrass Equation (Equation 1):(1)$$E:\;y^{2}+xy=x^{3}+ax+b$$
where a,b∈GF(p) and $4a^{3}+27b^{2}\neq 0\;\left(mod\;p\right)$. All points $\left(x,y\right)$ that satisfy Weierstrass Equation (Equation 1) and the infinite point *O* form an abelian group [21]. Let points $P=(x_{p},y_{p})$ and $Q=(x_{Q},y_{Q}$) be points on the elliptic curve. Elliptic curve cryptography defines two addition operations, point addition and point doubling. For P≠Q, the expression of PA is $R\left(x_{R},y_{R}\right)=P+Q$, and for P=Q, the expression of PD is $R\left(x_{R},y_{R}\right)=2P$.

The calculation formulas of PA and PD are different in different coordinate systems [21]. Table 1 lists the calculation cost of the $y^{2}+xy=x^{3}-3x+b$ curve in various coordinate systems. In Table 1, I denotes a modular inverse operation, M denotes a modular multiplication operation, and S represents a modular square operation. Since modular inversion is the most time-consuming of all basic operations, we use the projected coordinate system to eliminate modular inversion in PA and PD operations. The complexity of calculating PA using mixed Jacobin–affine coordinates is 8M + 3S, and the complexity of calculating PD using Jacobin coordinates is 4M + 4S. A modular square operation occupies the second important position in PA and PD operations. However, most previous architectures used one modular multiplication unit to calculate modular square operations. In this case, much of the PA and PD cycle time is wasted, resulting in a low performance level during the PM process. Therefore, it is necessary to design a modular square unit.

To switch between different coordinates, coordinate conversion is required. Equation (Equation 2) is for converting affine coordinates into Jacobin coordinates. Equation (Equation 3) is for converting Jacobin coordinates into affine coordinates.
(2)$$\left(x,y\right)\to \left\{\left(X,Y,Z\right)|X=x,Y=y,Z=1\right\}$$
(3)$$\left(X,Y,Z\right)\to \left\{\left(x,y\right)|x=X/Z^{2},\;y=Y/Z^{3}\right\}$$

In mixed Jacobian–affine coordinates, the calculation formulas of PA are:(4)$$\left\{\begin{array}{c} X_{3}={\left(Y_{2}{Z_{1}}^{3}-Y_{1}\right)}^{2}-{\left(X_{2}{Z_{1}}^{2}-X_{1}\right)}^{2}\left(X_{1}+X_{2}{Z_{1}}^{2}\right)\\ Y_{3}=\left(Y_{2}{Z_{1}}^{3}-Y_{1}\right)\left(X_{1}{\left(X_{2}{Z_{1}}^{2}-X_{1}\right)}^{2}-X_{3}\right)-Y_{1}{\left(X_{2}{Z_{1}}^{2}-X_{1}\right)}^{3}\\ Z_{3}=\left(X_{2}{Z_{1}}^{2}-X_{1}\right)Z_{1}. \end{array}\right.$$

In Jacobian coordinates, the calculation formulas of PD are:(5)$$\left\{\begin{array}{c} X_{3}={\left(3{X_{1}}^{2}+a{Z_{1}}^{4}\right)}^{2}-8X_{1}{Y_{1}}^{2}\\ Y_{3}=\left(3{X_{1}}^{2}+a{Z_{1}}^{4}\right)\left(4X_{1}{Y_{1}}^{2}-X_{3}\right)-8{Y_{1}}^{4}\\ Z_{3}=2Y_{1}Z_{1} \end{array}\right.$$

A point multiplication (PM) operation on the elliptic curve is defined as kP=(k/2)P+(k/2)P, where *k* is a positive integer. PM can be transformed into operations of PA and PD. The non-adjacent form (NAF) algorithm is a standard PM calculation method that can reduce the number of PM calculations to *k* PD and k/3 PA. The NAF encoding of *k* can be expressed as $k=\sum_{i=0}^{l-1}k_{i}2^{i}$, where $k_{i}\in 0,\pm 1,k_{l-1}\neq 0$, and no two consecutive digits $k_{i}$ are non-zero. NAF encoding has the fewest non-zero numbers.

However, when calculating PM with the NAF algorithm, it is essential to convert binary *k* into NAF encoding in advance, which wastes many clock cycles. Algorithm 1 can avoid the conversion of NAF encoding by pre-calculating h=3k, and the cost is only an extra register resource. Algorithm 1 simultaneously scans the values of $h_{i}$ and $k_{i}$ from left to right, performing PD once per scan bit, performing PA once when $h_{i}=1$ and $k_{i}=0$, and performing point subtraction (PS) once when $h_{i}=0$ and $k_{i}=1$. Since $Q-P=Q+\left(-P\right)$, where $-P=\left(x,-y\right)$ for P=(x,y), PS can be converted into PA with almost no cost. This design uses Algorithm 1 to calculate the PM.
**Algorithm 1** On-the-fly NAF method for PM**Input**: $k=\left(k_{n-1},k_{n-2},\cdots ,k_{1},k_{0}\right)$, $P=\left(x,y\right)$**Output**: kP1: $h=3k=\left(h_{l-1},h_{l-2},\cdots ,h_{1},h_{0}\right)$,where $h_{l}$ is 12: Q←O3: **For** *i* from l−1 down to 1 **do**    3.1: Q←2Q    3.2: **if** $\left(h_{i}=1\right)$ and $\left(k_{i}=0\right)$, **then** Q←Q+P    3.3: **if** $\left(h_{i}=0\right)$ and $\left(k_{i}=1\right)$, **then** Q←Q−P4: **return** *Q*.

### 2.2. Modular Multiplier Algorithm

Modular multipliers consume most of the hardware resources in the point multiplication architecture and dominate the overall performance. The Montgomery algorithm uses a series of low-cost addition and shift operations instead of modular operations to reduce computational complexity. However, large-number multiplication is still challenging for hardware. Therefore, a radix-$2^{w}$ Montgomery modular multiplication algorithm is proposed for reducing the multiplication of large numbers into a shorter set of integer multiplications. As shown in Algorithm 2, *k*-bits multiplier *a* is denoted as $a=(a_{\left(s-1\right)},a_{\left(s-2\right)},\ldots ,a_{0})$, where $a_{i}$ is w-bit data. $p_{0}^{{}^{\prime }}$ is the precomputed parameter that can be calculated with the $r\cdot r^{{}^{\prime }}-p_{0}^{{}^{\prime }}\cdot p=1$ formula. It takes s/w cycles to perform one Montgomery modular multiplication. Steps 2.2 to 2.4 are equivalent to computing a reduction in $R\left(x\right)=x\cdot 2^{-w}mod\;p$ once.
**Algorithm 2** Montgomery multiplication algorithm**Input**: $a=\left(a_{s-1},a_{s-2},\ldots ,a0\right)$, *b*, *p* with $a,b\;\in \left[0,\;p\right)$**Output**: $Monpro\left(a,b\right)=a\cdot b\cdot r^{-1}mod\;p$1: u=0.2: **For** *i* from 0 up to s−1 **do**    2.1: $u=u+a_{i}\cdot b$    2.2: $m=u_{0}\cdot p_{0}^{\prime }\;mod\;2^{w}$    2.3: u=u+m·p    2.4: $u=u/2^{w}$3: **if** u≥p, **then return**u−p                    **else return** *u*

The Montgomery algorithm can be applied to most prime number fields, but rarely to NIST prime number fields because NIST prime fields have unique modular reduction methods. A Montgomery fast partial product reduction algorithm was proposed in [17] that combines the Montgomery algorithm with NIST prime fields. The advantage of this algorithm is that it avoids the two multiplications in Steps 2.2 to 2.4.

The Montgomery fast partial product reduction algorithm denotes *x* as $x=c\cdot 2^{w}+l$; then,
(6)$$\begin{array}{cc} R(x) & =\left(c\cdot 2^{w}+l\right)*2^{-w}mod\;p\\ \; & =(c+l\cdot ((p+1)>>w))mod\;p\\ \; & =(c+r)mod\;p \end{array}$$
when $p=p_{256}=2^{256}-2^{224}+2^{192}+2^{96}-1$ and w<96:(7)$$\begin{array}{cc} r & =l\cdot (\left(p_{256}+1\right)>>w)\\ \; & =l\cdot (\left(2^{256}-2^{224}+2^{192}+2^{96}\right)>>w)\\ \; & =(l<<(256-w))-(l<<(224-w))+(l<<(192-w))+(l<<(96-w)) \end{array}$$

Therefore, after deriving the value of *r* via simple addition and shift operations, the reduction result of $R\left(x\right)=x\cdot 2^{-w}mod\;p$ can be easily obtained.

Compared with the traditional algorithms, the Montgomery fast partial product reduction algorithm speeds up modular multiplication computation and is more hardware-friendly. However, considering the high-speed requirements for encryption and decryption, the performance of a single modular multiplication module is still limited. Modular multiplication and modular square operations must be computed in parallel to further improve the speed of PM operation.

## 3. Implementations

In this section, we propose a high-performance PM architecture that calculates modular multiplication and square operations in parallel. Secondl the algorithms and circuits of modular multiplication and modular square are introduced. Then, a new high-speed modular inverse circuit is proposed by improving the modular inverse algorithm. Lastly, we analyze data dependency in the PA and PD operations, and perform calculations to achieve high speed.

### 3.1. Proposed PM Architecture

The proposed PM architecture is shown in Figure 1. It consists of one modular square unit, one modular multiplication unit, two modular adder units, one main state machine, and several data registers. Compared to the one modular multiplication unit used in traditional PM architectures, this novel architecture utilizes one more modular square unit to achieve higher speed. The cost of the area is not high, but better efficiency is achieved.

The proposed PM state machine is shown in Figure 2. At the beginning of the PM calculation, input coordinates $P=(x_{p},y_{p})$ are stored in x2, y2 registers, and the *k* integer is stored in the *k* register. Second, the *h* register is used to store the value of h=k+k·2. Then, coordinate data are converted into Montgomery field numbers and Jacobin coordinates. The control unit cyclically calculates PA and PD on the basis of *k* and *h*, and outputs intermediate variable *Q* in the x3, y3, and z3 registers. In the end, output result $Q=kP=(x_{Q},y_{Q})$ is obtained after finishing modular inversion and final coordinate transformation.

### 3.2. Modular Multiplication Unit and Modular Square Unit

To improve the speed of modular multiplication and to be more friendly to hardware implementation, the Montgomery fast partial product reduction algorithm for modular multiplication is considered in this paper. As shown in Algorithm 3, the proposed modular multiplication algorithm is executed once in $Monpromul\left(a,b\right)=a\cdot b\cdot r^{-1}mod\;p$ operation. Algorithm 3 has three functions: multiplication, reduction, and modular addition, each performed by a separate circuit module. For *k*-bit data *a* and *b*, it takes s=k/w iterations of $a_{i}\cdot b$ multiplication, *s* times reduction, and final modular addition. In this paper, we describe the design for k=256 and w=64. Because CSA adders are used instead of a part of the adder in Algorithm 3, there are pairs of variables. The CSA adder helps in reducing the circuit’s critical path, but two values represent the result.
**Algorithm 3** Proposed Montgomery multiplication algorithm**Input**: $a=\left(a_{3},a_{2},a_{1},a_{0}\right)$, *b*, *p* with $a,b\;\in \left[0,\;p\right)$**Output**: $Monmul\left(a,b\right)=a\cdot b\cdot r^{-1}mod\;p$1: $\left(c0,c1\right)$ ← $Mul(a_{0},b)$2: $\left(c0,c1\right)$ ← $Mul(a_{1},b)$   (d0,d1) ← Reduction(c0,c1,0,0)3: **For** *i* from 2 up to s−1 **do**   $\left(c0,c1\right)$ ← $Mul(a_{i},b)$   (d0,d1) ← Reduction(c0,c1,d0,d1)4: (d0,d1) ←Reduction(c0,c1,d0,d1)5: *c* ← Add(d0,d1)

Since the resources required for modular square are similar to modular multiplication, it is most important to organize it well to achieve better efficiency. The proposed modular square algorithm is based on the same idea as that of the proposed modular multiplication algorithm. Therefore, the modular square can benefit from high-speed modular reduction circuits. As shown in Algorithm 4, the proposed modular square algorithm is executed once $Monpromul\left(a\right)=a^{2}\cdot r^{-1}mod\;p$ operation. Algorithm 4 has three functions: square, reduction, and modular addition, each performed by a separate circuit module. For *k*-bit data *a* and *b*, it takes s=k/w iterations of $a_{i}\cdot b$ square, *s* times reduction, and final modular addition. The difference between the modular square algorithm and the modular multiplication algorithm is mainly in the circuit structure of the square and reduction operation.
**Algorithm 4** Proposed Montgomery square algorithm**Input**: *a*, *p* with $a\in \left[0,\;p\right)$**Output**: $Monsqu\left(a\right)=a^{2}\cdot r^{-1}mod\;p$1: $\left(c2,c3\right)$←Squ(a,0)2: $\left(c2,c3\right)$ ← Squ(a,1)   (d2,d3) ← Reduction(c2,c3,0,0)3: **For** *i* from 2 up to s−1 **do**   $\left(c2,c3\right)$ ← Squ(a,i)   (d2,d3) ← Reduction(c2,c3,d2,d3)4: (d2,d3) ←Reduction(c2,c3,d2,d3)5: *c* ← Add(d2,d3)

Figure 3 shows the circuit structures of the modular multiplication unit and the modular square unit. The *mul* module performs the calculation of a·b and is highlighted in green, while the *squ* module performs the calculation of $a^{2}$ and is also highlighted in green. The red1 and red2 modules, highlighted in blue, are used for the reductions in Algorithms 3 and 4, respectively. The yellow-marked modules add1 and add2 are two identical modular adder circuits that can be operated independently. The mul_a and mul_b registers are used to store the multiplicands *a* and *b* in Algorithm 3, while the squ_a register is used to store the square data *a* in Algorithm 4. The add1_a and add1_b registers store the input data for the add1 circuit, and the add2_a and add2_b registers store the input data for the add2 circuit.

In Algorithm 3, since reduction always follows multiplication, Step 1 does not perform the reduction, and Step 4 does not perform the multiplication. This renders the pipelined structure possible. As shown in Figure 4, both the modular multiplication and modular squaring circuits have a three-stage pipeline structure. It takes only 4 clock cycles to perform a modular multiplication or a modular square operation in a sequential calculation.

In Algorithm 3, with four 64-bit multipliers, one 256-bit × 64-bit multiplication operation is executed in 4 different clock cycles. Every clock cycle time, the four 64-bit multiplications results are combined and stored in registers c0 and c1, denoted as $mul\left(a_{i},b\right)=c0+c1\cdot 2^{64}$. Figure 5 shows the calculation of c0 and c1 with different values of i. In Algorithm 3, the values of c0 and c1 in each cycle were 64 bits higher than those of the last cycle. On the basis of this characteristic, we designed the squ function in Algorithm 4.

In Algorithm 4, with nine 32-bit multipliers, one 256-bit square operation is executed in 4 different clock cycles. Every clock cycle time the nine 32-bit multiplications results are combined and stored in registers c2 and c3, expressed as $squ\left(a,i\right)=c2+c3\cdot 2^{64}$. Figure 6 and Figure 7 illustrate the calculation of c2 and c3 for different values of *i*. Similarly, in Algorithm 4, the values of c2 and c3 in each cycle were 64 bits higher than those in the last cycle. This characteristic makes it possible to design the high-speed modular reduction unit for modular square. In addition, compared to the four 64-bit multipliers in the modular multiplication unit, the modular square unit uses only nine 32-bit multipliers, resulting in a significant reduction in area.

In the red1 module, CSA257 and CSA258 are used to calculate c0+c1+d0+d1 to obtain the results of both result1 and carry1. As shown in Figure 8, a 64-bit adder adds the lower 64-bit of result1 and carry1 to get the value of $h\cdot 2^{64}+l$. The calculation of *r* is in Equation (Equation 7), and the specific implementation is shown in Figure 9. The value of *r* is obtained using a 96-bit adder after shifting *l* left and merging it. The outputs of CSA256 and carry1 are strobed back to red1’s inputs, ensuring that the length of the critical path is not affected.

As shown in Figure 3, red2 had a similar structure to that of red1, but some modifications were performed for different bit widths. When Algorithm 4 loops to Step 4, the value of the high 65-bit of c2 is $a7\cdot a7\leq \left(2^{32}-1\right)\times \left(2^{32}-1\right)=\left(2^{64}-2^{33}+1\right)$, so $c2<2^{320}-2^{289}+2^{256}+2^{256}-1$. After the data pass through CSA258 and CSA289, the value of *u* is $u=u+squ\left(a\right)<2^{320}-2^{288}-2^{287}$. Then, through the 64-bit shift register, the value of *c* is $c<2^{256}-2^{224}-2^{223}<p_{256}$. Lastly, at CSA257, because $r=\left(l\ll 192\right)-\left(l\;\ll 160\right)+\left(l\;\ll 128\right)+\left(l\;\ll 32\right)<p_{256}$, then $c+r<2\cdot p_{256}$, the output of the red2 module can be connected to the input of the modular adder units.

### 3.3. Modular Inversion Unit

Modular inversion operation is required for coordinate conversion. Modular inversion can be implemented by using the extended Euclidean algorithm. The extended Euclidean algorithm comprises the Radix-2 and Radix-4 algorithms [22,23,24,25]. The Radix-2 algorithm is particularly useful for hardware implementation, and the modular inverse result can be obtained by circularly performing subtraction and shift operations. The authors in [24] showed that the Radix-2 algorithm requires 363 clock cycles to calculate a modular inverse operation on average. Compared with the Radix-2 algorithm, the Radix-4 algorithm has changed the number of bits per cycle scan from 1-bit to 2-bit, and its speed could theoretically be twice that of the Radix-2 algorithm. However, the Radix-4 algorithm needs to consume a lot of circuit resources and is not suitable for hardware. In this paper, we propose a a high-speed modular inversion algorithm based on the Radix-2 and Radix-4 algorithms. It takes about 300 clock cycles to execute one modular inversion operation, which is only 83% of the Radix-2 algorithm.

Algorithm 5 implements the division by 4 and division by 2 operations of *u* and *v* by right shifting. During the division by 4 and by 2 operations of x1 and x2, the least significant bits of the respective values must be set to zero by using the +0, +P, +2P, and +3P operations, followed by a right shift to obtain the result. The modular inverse unit uses two adders to update the values of registers *u* and *v* in circuit design, and uses two modular adders to update the values of registers x1 and x2. To minimize the area of the ECC circuit, registers *u* and *v* are multiplexed with registers storing the affine coordinate input data $P=(x_{p},y_{p})$, and registers x1 and x2 are multiplexed with the modular addition unit input registers.

As shown in Figure 10, the hardware structure of the modular inversion module consists of two adders in (a) and (b), and two modular addition modules in (c) and (d). Within the elliptic curve point multiplication circuit, the modular addition module serves as a shared arithmetic resource. On the one hand, it can be used for modular addition and subtraction operations in point addition and point doubling calculations. On the other hand, it also assists in implementing modular inversion, multiplication, and squaring operations. To achieve diverse functionality, the modular addition module incorporates several data selectors and shift registers, building on the existing two 257-bit adders, and is designed with four output ports.
**Algorithm 5** Proposed modular inversion algorithm**Input**: *a*, *p* with $a\in \left[0,\;p\right)$**Output**: $a^{-1}mod\;p$1: u=a,v=p,x1=1,x2=02: **while** ((u!=1)&(v!=1))     2.1: **if** u[1:0]=2’b00          u=u/4,x1=x1/4modp     2.2: **else if** v[1:0]=2’b00          v=v/4,x2=x2/4modp     2.3: **else if** u[0]=1’b0          u=u/2,x1=x1/2modp     2.4: **else if** v[0]=1’b0          v=v/2,x2=x2/2modp     2.5: **else if** u≥v          u=(u−v)/2,x1=(x1−x2)/2modp     2.6: **else**          v=(v−u)/2,x2=(x2−x1)/2modp3: **if** u=1, **return**x1   **else      return** x2

### 3.4. Calculation of PA and PD

In the circuit implementation of PA and PD operations, due to the strong data dependency in the calculation formula sequence, hardware utilization cannot reach 100%. To improve the computation speed and hardware resource utilization of the system, this section first analyzes the data flow of PA and PD operations. Then, we decompose them into a series of finite field operation steps, and re-schedule and coordinate the usage of arithmetic logic units. The computation sequences for point addition and point multiplication operations are illustrated in Figure 11 and Figure 12, respectively.

Figure 11 shows that, during PA operation, State 1 only executes the modular squaring operation, leaving the modular multiplication unit idle. Meanwhile, in States 6–8, only modular multiplication and modular addition operations are executed, rendering the modular squaring unit idle. Similarly, Figure 12 shows that, during PD operation, State 1 only executes the modular squaring operation, leaving the modular multiplication unit idle; while in States 5 and 6, only modular multiplication and modular addition operations are executed, rendering the modular squaring unit idle.

To enhance hardware resource utilization and shorten computation time, employing pipeline technology is an effective method for addressing idle time issues between modular multipliers and modular squarers. Pipeline technology divides the calculation process of PA and PD into multiple stages, allowing tasks from different stages to be executed simultaneously. By adopting a neighboring task connection strategy, gaps between modular multipliers and modular squarers are effectively filled. As a result, the next PA or PD operation can be initiated immediately before the completion of a previous one, thereby achieving efficient parallel computation. It takes 21 cycles to compute a PD, and 32 cycles to compute a PA. The number of operation clock cycles is reduced to the minimum.

## 4. Results and Comparisons

As shown in Figure 13, the proposed architecture was implemented on Xilinx Zynq-7000 FPGA by Vivado 2018. The synthesized results on Virtex-7 FPGA compared with those of other works are shown in Table 2. The elliptic curve was selected from the 256-bit NIST-$p_{256}$ curve or SM2 curve. The performance of the different designs was evaluated via the product of implementation area (LUTs) and the time (ms). To better evaluate the trade-off between area and speed in digital circuits, the AT parameter (AT = area (LUTs) × time (ms)) is introduced.

The proposed design on Virtex-7 FPGA reached the frequency of 105.3 MHz and performed one PM operation in 0.08 ms at the cost of 23.1k LUTs. Having achieved an AT of 1.85, the proposed architecture achieved relatively balanced performance with low latency and acceptable regional costs. Compared to [18], even with the smallest area design of 16.3k LUTs, our speed was more than 34 times faster than theirs, and our frequency was higher. In [26], the proposed architecture employs a dual-core hardware design along with a radix-128 Montgomery modular multiplication algorithm to attain remarkably high speeds. Although it executes one PM in a mere 0.056 ms, it needs 182k LUTs, which is 7.8 times more than that of our approach. The authors in [27] presented a well-balanced solution for performance and resource utilization that takes only 0.15 ms to execute one PM, reaches a frequency of 123.3 MHz, and has an area of just 22.9k LUTs. The comprehensive AT value was 3.44, which was 1.86 times more than that in our work. Our proposed design calculates modular multiplication and square operations in parallel, delivering the best AT performance among existing works. Consequently, this design is exceptionally well-suited for high-speed encryption and decryption in IoT applications.

## 5. Conclusions

This paper presented a high-efficiency and low-delay ECC architecture based on NIST-$p_{256}$ prime field for IoT security applications. A modular square unit was designed using fast partial Montgomery reduction algorithm that takes only four clock cycles to complete a modular square operation. The modular multiplication and modular square operations can be computed in parallel to achieve high speed. The proposed design on Virtex-7 FPGA reached a frequency of 105.3 MHz and performed one PM operation in 0.08 ms at the cost of 23.1k LUTs. Having achieved an AT of 1.85, the proposed architecture achieved relatively balanced performance with the lowest latency and acceptable regional costs. Consequently, this design is exceptionally well-suited for high-speed encryption and decryption in IoT applications.

## Figures and Tables

**Figure 1 micromachines-14-01037-f001:**
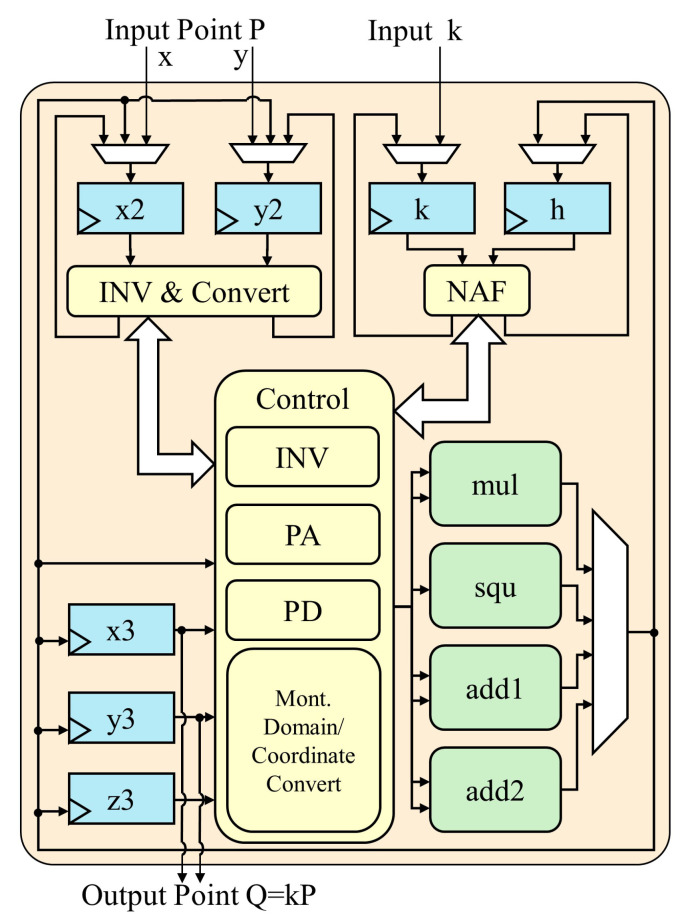
Proposed PM architecture.

**Figure 2 micromachines-14-01037-f002:**
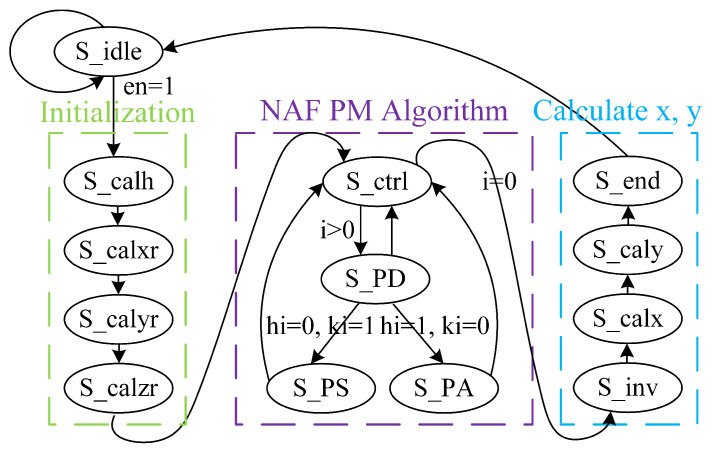
Proposed PM state machine.

**Figure 3 micromachines-14-01037-f003:**
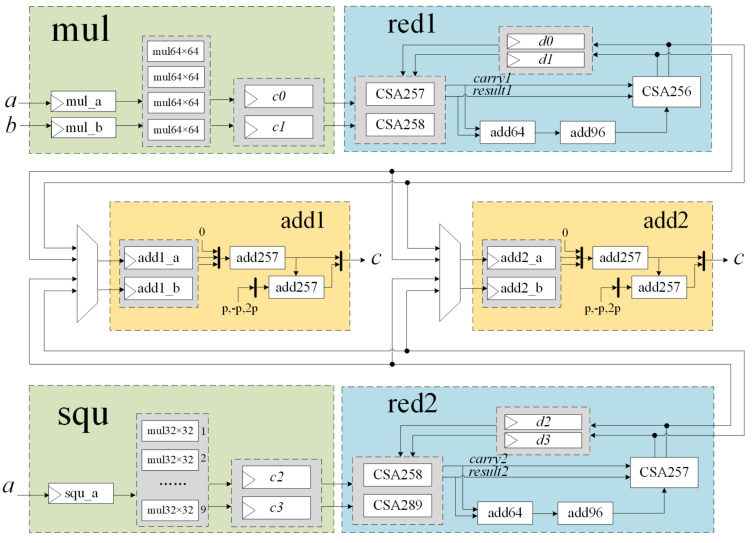
Structure of the modular multiplication unit and the modular square unit.

**Figure 4 micromachines-14-01037-f004:**
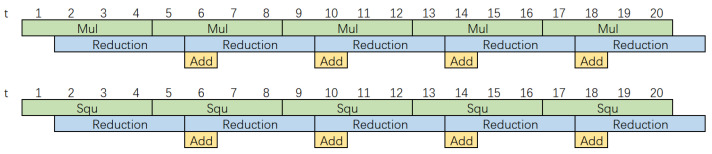
Pipeline of the modular multiplication unit and the modular square unit.

**Figure 5 micromachines-14-01037-f005:**
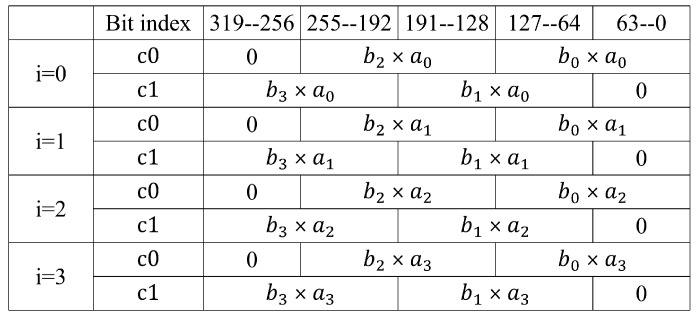
Calculation of c0 and c1.

**Figure 6 micromachines-14-01037-f006:**
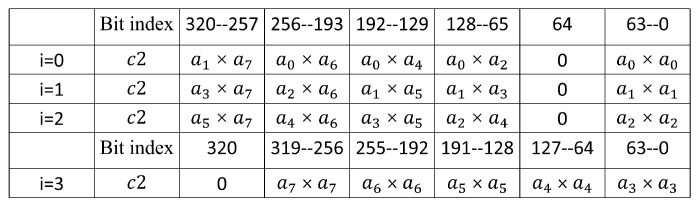
Calculation of c2.

**Figure 7 micromachines-14-01037-f007:**
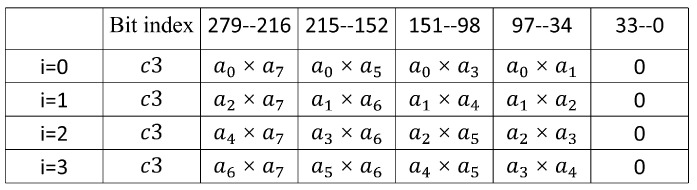
Calculation of c3.

**Figure 8 micromachines-14-01037-f008:**
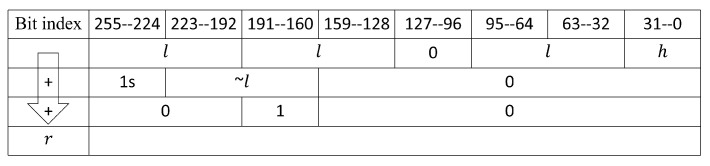
Calculation of *l*.

**Figure 9 micromachines-14-01037-f009:**
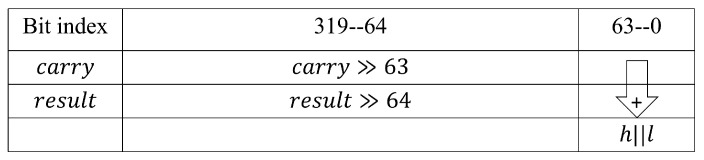
Calculation of *r*.

**Figure 10 micromachines-14-01037-f010:**
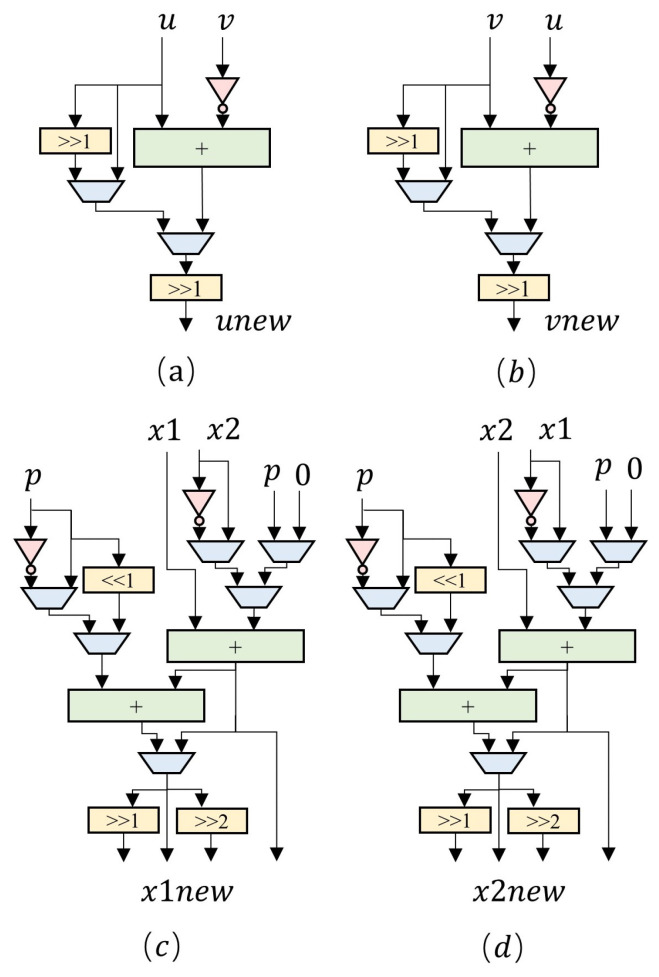
Structure of adders and modular adders.

**Figure 11 micromachines-14-01037-f011:**
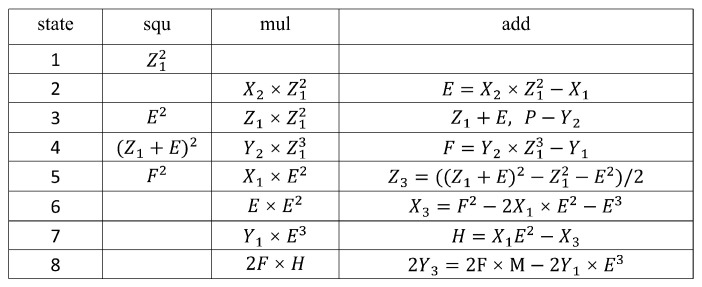
Calculation of PA.

**Figure 12 micromachines-14-01037-f012:**
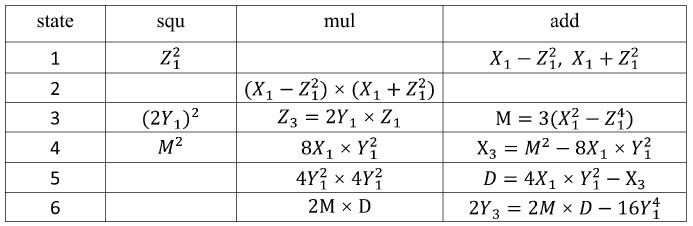
Calculation of PD.

**Figure 13 micromachines-14-01037-f013:**
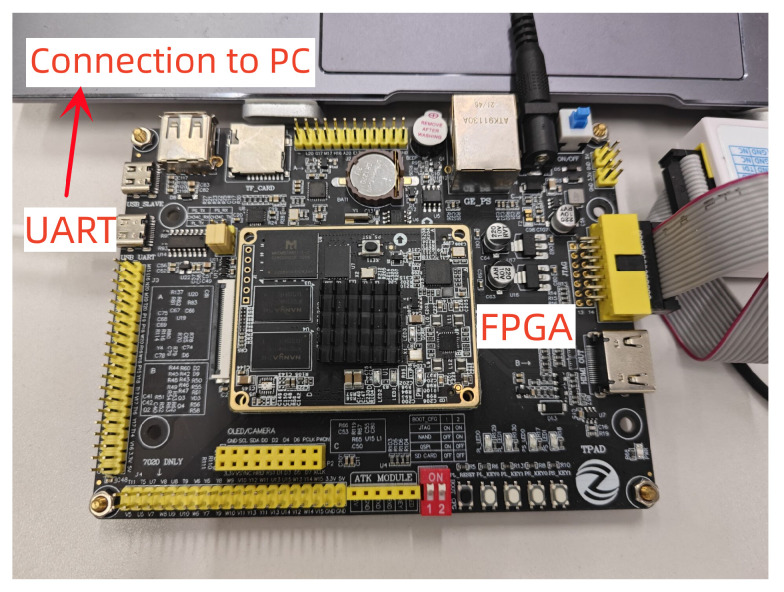
FPGA test platform.

**Table 1 micromachines-14-01037-t001:** Calculation cost of PA and PD.

Coordinate	PA	PD
Affine coordinate	1I ${}^{1}$ + 2M ${}^{2}$ + 1S ${}^{3}$	1I + 2M + 2S
Projective coordinate	12M + 2S	7M + 3S
Jacobin coordinate	12M + 4S	4M + 4S
Chudnovsky coordinate	11M + 3S	5M + 4S
Mixed Jacobin-affine coordinate	8M + 3S	

^1^ I, modular inversion operation; ^2^ M, modular multiplication operation; ^3^ S, modular square operation.

**Table 2 micromachines-14-01037-t002:** Comparison with other PM architectures on FPGA.

Works	FPGA	Area/LUTs	Freq./MHz	Time/ms	AT
[18]	virtex-4	16.3k	62.4	2.75	44.83
[14]	Virtex-5	26.7k	34.15	0.42	11.21
[26]	Virtex-6	182k	104	0.056	10.19
[16]	Virtex-6	18.6k	121.6	0.3	5.58
[27]	Virtex-7	22.9k	123.3	0.15	3.44
Ours	Virtex-7	23.1k	105.3	0.08	1.85

## Data Availability

Not applicable.

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
