# Peer review of "FPGA Implementation for Elliptic Curve Cryptography Algorithm and Circuit with High Efficiency and Low Delay for IoT Applications"

_micromachines, 2023, doi:10.3390/mi14051037_

Round 1

Reviewer 1 Report

The article is about the FPGA Implementation for ECC.

The topic is interesting and current. The text is well-written, and the manuscript is easy to read. However, the following remarks will improve the quality of the article:

1) There are typos in the text, e.g., "2. preliminaries"

2) Some expressions are not in math mode, e.g., E above formula (1); more similar errors are later in the text.

3) Page 2 last paragraph: should be Table 1

4) I suggest adding a list of used abbreviations in the form of a table. PA or PD sounds very mysterious in the text.

5) some notations are inconsistently used, e.g., p_{256} and p256

6) I suggest the names of the variables used in the algorithms to distinguish in the text. x2 or red1 merge with other words in sentences.

7) Related work can be extended with other thematically related references, e.g.

https://doi.org/10.3390/electronics10111252

DOI: 10.1109/WSCNIS.2015.7368282

Reviewer 2 Report

Dear Authors,

 The content of your article fits perfectly in the scope of the special issue "FPGA Applications and Future Trends" of Micromachines journal. One reason for this is that a key study topic involves the issue of Internet of Things (IoT) security. This is very important in terms of solutions meeting the rapidly increasing requirements to guarantee an adequate level of IoT security in terms of smart terminal devices authentication, software vulnerabilities, and privacy leaks.

The authors focused their research on the hardware implementation of an elliptic curve cryptography (ECC) architecture well suited for use in IoT security. It is relevant and interesting as the manuscript describes a new, previously unpublished contribution to the field under consideration.

This was based on the 19 publications analysed in the two initial sections of the article and representing the most relevant advances in the field by the authors. The references are relevant to the topic and cover both historical literature and more recent developments.

The methodology presented and its principles in the third section are reasonable and appropriate.

The results of an effective analysis of the cybersecurity under consideration are presented and explained.

The results of an effective comparative analysis of the performance of a field-programmable gate array (FPGA) structure as a hardware implementation with other works are presented and explained, which is reflected in the fourth section.

The authors proved that the results they obtained offer a relatively balanced performance with the lowest latency and acceptable regional costs, this consequently means , the proposed design is exceptionally well-suited for high-speed encryption and decryption in IoT applications.

The paper contains some new data.

The paper is presented in logical way and overall written well.

All notations and terminology are clearly defined.

The content of the Conclusion is consistent with the evidences and arguments presented and addresses the main question asked.

 Comments and Suggestions for Authors

1.      It would be best to clearly identify which of the previous works by authors constitute the foundation of the work presented in this article.

2.      The study considered the NIST-P256 primary field for IoT security applications. What other alternatives were there?

 Best regards,

Reviewer
